# MicroRNA-Mediated Responses to Cadmium Stress in *Arabidopsis thaliana*

**DOI:** 10.3390/plants10010130

**Published:** 2021-01-10

**Authors:** Joseph L. Pegler, Jackson M. J. Oultram, Duc Quan Nguyen, Christopher P. L. Grof, Andrew L. Eamens

**Affiliations:** 1Centre For Plant Science, School of Environmental and Life Sciences, Faculty of Science, University of Newcastle, Callaghan, NSW 2308, Australia; joseph.pegler@newcastle.edu.au (J.L.P.); Jackson.Oultram@uon.edu.au (J.M.J.O.); DucQuan.Nguyen@uon.edu.au (D.Q.N.); chris.grof@newcastle.edu.au (C.P.L.G.); 2Institute of Genome Research, Vietnam Academy of Research and Technology, 18 Hoang Quoc Viet Str., Cau Giay, Hanoi 100000, Vietnam

**Keywords:** *Arabidopsis thaliana* (*Arabidopsis*), cadmium (Cd) stress, microRNA (miRNA), miRNA-directed gene expression regulation, double-stranded RNA binding (DRB) protein, RT-qPCR

## Abstract

In recent decades, the presence of cadmium (Cd) in the environment has increased significantly due to anthropogenic activities. Cd is taken up from the soil by plant roots for its subsequent translocation to shoots. However, Cd is a non-essential heavy metal and is therefore toxic to plants when it over-accumulates. MicroRNA (miRNA)-directed gene expression regulation is central to the response of a plant to Cd stress. Here, we document the miRNA-directed response of wild-type *Arabidopsis thaliana* (*Arabidopsis*) plants and the *drb1*, *drb2* and *drb4* mutant lines to Cd stress. Phenotypic and physiological analyses revealed the *drb1* mutant to display the highest degree of tolerance to the imposed stress while the *drb2* mutant was the most sensitive. RT-qPCR-based molecular profiling of miRNA abundance and miRNA target gene expression revealed DRB1 to be the primary double-stranded RNA binding (DRB) protein required for the production of six of the seven Cd-responsive miRNAs analyzed. However, DRB2, and not DRB1, was determined to be required for miR396 production. RT-qPCR further inferred that transcript cleavage was the RNA silencing mechanism directed by each assessed miRNA to control miRNA target gene expression. Taken together, the results presented here reveal the complexity of the miRNA-directed molecular response of *Arabidopsis* to Cd stress.

## 1. Introduction

In plants, heavy metals such as copper (Cu), zinc (Zn) and iron (Fe) are essential elements required for numerous physiological and biological processes ensuring normal growth and development, whereas cadmium (Cd), mercury (Hg) and aluminum (Al) are non-essential heavy metals, and therefore, become toxic to the plant when they over-accumulate [1,2,3]. Cd is a natural component of the earth’s crust and is routinely released into the environment from volcanic events, rock weathering and via soil erosion [4,5]. However, the abundance of environmental Cd has increased rapidly in recent decades due to various agricultural and industrial practices [6,7,8]. Furthermore, due to the high mobility of Cd ions in soil, they are readily taken up by plant root systems and subsequently translocated to the aerial tissues, where Cd ions can over accumulate in edible organs such as leaves, grains and fruits [2,7,9,10]. As a non-essential heavy metal, Cd provides no beneficial biological function in humans; therefore, contamination of our food chain with Cd poses serious health concerns, including chronic toxicity to the kidneys, bones and lungs [7,10,11,12].

The over-accumulation of Cd in plant cells affects various physiological and biochemical processes, primarily via its disruption of enzymatic-based systems [1,2,4]. Cd ions rapidly bind both apoplastic and symplastic proteins to disrupt their activity and induce a state of oxidative stress [1,2,4]. For example, plants exposed to elevated Cd display an array of toxicity symptoms in their above- and below-ground organs. More specifically, in aerial tissues, Cd stress decreases chlorophyll abundance, damages the photosynthetic apparatus, alters respiration rates and causes water imbalances [7,9,12]. In the roots, Cd stress primarily inhibits water acquisition and prohibits the uptake of essential nutrients [11,13,14]. Phenotypically, inhibition of primary root elongation and the repression of lateral root formation are the most commonly reported consequences of Cd stress [1,3,14]. Plants do not encode specific transporter proteins for the uptake of Cd from the soil by the roots, nor for its translocation to the shoot [5,9,10,14]. However, the similar physicochemical properties of Cd ions to Cu, Fe and Zn ions allows Cd to enter root cells for its subsequent efflux into the xylem by a range of transporter proteins, including members of the heavy metal ATPase (HMA), ATP-binding cassette (ABC), natural resistance-associated macrophage protein (NRAMP) and Fe superoxide dismutase (FSD) protein families [2,3,10,14,15,16,17]. Evidence of the involvement of a wide range of transporter proteins in Cd uptake and transport primarily stems from their altered expression upon the imposition of Cd stress [2,3,7,10].

In order to appropriately respond to heavy metal stress via the control of the uptake, efflux, translocation and sequestration of heavy metal ions, plants have evolved complex, multilayered regulatory mechanisms. At the post-transcriptional level, heavy metal stress responses have been repeatedly demonstrated to primarily be in the form of microRNA (miRNA)-directed gene expression regulation, with the highly conserved miRNAs miR160, miR167, miR393, miR395, miR396, miR399 and miR408 identified in *Arabidopsis*, canola (*Brassica napus*), maize (*Zea mays*), barrelclover (*Medicago truncatula*), alfalfa (*Medicago sativa*), rice (*Oryza sativa*), radish (*Raphanus sativus*), soybean (*Glycine max*), sunflower (*Helianthus annuus*) and wheat (*Triticum aestivum*) to be responsive to Cd stress [2,4,6,7,8,9,12,14,15,18]. Furthermore, the identification of miRNAs specific to the auxin pathway (miR160, miR167 and miR393), in addition to the identification of the sulfur (S), phosphate (PO_4_) and Cu stress-responsive miRNAs, miR395, miR399 and miR408, respectively, infers the complexity of a potentially “shared” or “common” miRNA-directed molecular response to Cd stress across a diverse array of evolutionary distantly related plant species [1,2,3,6,7,9,11,12,18].

*Arabidopsis* encodes five double-stranded RNA binding (DRB) proteins with three nucleus-localized DRBs, DRB1, DRB2 and DRB4, each assigned a functional role in the miRNA pathway [19,20,21,22,23,24,25,26,27]. More specifically, DRB1 and DRB4 have been demonstrated to functionally interact with the Dicer-like (DCL) endonucleases DCL1 and DCL4, respectively [19,20], with DRB1 and DCL1 shown to be required for the production of conserved miRNAs [19,21,22], and DRB4 and DCL4 to be involved in the production of newly evolved miRNAs [20,23]. Unlike DRB1 and DRB4, DRB2 appears to be able to functionally interact with either DCL1 or DCL4 for the production of conserved and newly evolved miRNAs, respectively [24,25]. In addition, DRB2 and DRB1 have been suggested to function as “molecular switches” to determine whether an *Arabidopsis* miRNA will regulate target gene expression via the canonical transcript cleavage mechanism of miRNA-directed RNA silencing or via the alternate mechanism of miRNA target gene expression regulation in plants, translational repression [26,27].

Therefore, here we report on the exposure of 8-day-old wild-type *Arabidopsis* plants (ecotype; Columbia-0 (Col-0)) and the *drb1*, *drb2* and *drb4* mutant lines to a 7-day 50 μM CdCl_2_ stress treatment regime to determine the requirement of either DRB1, DRB2 or DRB4 for *Arabidopsis* to mount a miRNA-directed molecular response to Cd stress. When taken together, the phenotypic and physiological assessments of Cd-stressed Col-0, *drb1*, *drb2* and *drb4* seedlings revealed that the *drb1* mutant displayed the highest degree of tolerance to Cd stress and that the *drb2* mutant was the most sensitive to the imposed stress. Furthermore, quantitative reverse transcriptase polymerase chain reaction (RT-qPCR) profiling of the abundance of seven miRNAs revealed that DRB1 was the primary DRB protein required for the production of six of the assessed miRNAs, including miR160, miR167, miR393, miR395, miR399 and miR408. However, both DRB2 and DRB4 were identified to play secondary roles in regulating the production of each of these miRNAs. For the seventh miRNA analyzed in this study, miR396, DRB2 was revealed to be the primary DRB protein required for its production. However, DRB1 and DRB4 were demonstrated to play important secondary roles in this process. Although both DRB1 and DRB2 were demonstrated to be required for Cd-responsive miRNA production, target transcript cleavage was revealed to be the predominant mode of RNA silencing directed by the seven analyzed miRNAs to regulate target gene expression. When taken together with the previous demonstration that each of the seven analyzed miRNAs are responsive to other forms of abiotic and/or biotic stress in *Arabidopsis*, and when considering the functional diversity of the proteins encoded by the target genes of this miRNA cohort, the results presented here provide additional insight into the extremely complex molecular response of *Arabidopsis* in an attempt to adapt to Cd stress.

## 2. Results

### 2.1. Assessment of the Phenotypic and Physiological Consequences of a 7-Day Cadmium Stress Treatment Regime on Arabidopsis Development

The severe developmental phenotype displayed by the *drb1* mutant is readily apparent by 15 days of age and is characterized by an overall reduced size compared to non-stressed (Ns) Col-0 (Col-0/Ns) seedlings (Figure 1A). More specifically, compared to the metrics of 21.3 mm^2^, 56.3 mm and 31.0 mg for the rosette area (Figure 1B), primary root length (Figure 1C) and fresh weight (Figure 1D) of 15-day-old Col-0/Ns seedlings, respectively, the corresponding metrics for non-stressed *drb1* (*drb1*/Ns) seedlings are reduced by 46.7%, 68.5% and 58.7%, respectively. Figure 1A also clearly presents the comparatively mild wild-type-like phenotypes displayed by non-stressed *drb2* and *drb4* (*drb2*/Ns and *drb4*/Ns) seedlings. Compared to Col-0/Ns plants, the rosette area, primary root length and fresh weight of *drb2*/Ns plants are mildly elevated by 4.2%, 2.1% and 19.0%, respectively (Figure 1B–D). Similarly, the rosette area, primary root length and fresh weight of *drb4*/Ns plants are mildly altered by −4.5%, +3.9% and −11.6%, respectively (Figure 1B–D).

The 7-day cultivation period in the presence of 50 μM CdCl_2_ clearly induced a stress response in all four assessed *Arabidopsis* lines (Figure 1A). In Cd-stressed Col-0 (Col-0/Cd) plants for example, the rosette area (Figure 1B), primary root length (Figure 1C) and fresh weight (Figure 1D) were reduced by 34.3%, 18.5% and 36.5% respectively, compared to Col-0/Ns plants. Similarly, reductions of 39.2%, 25.7% and 41.7% were determined for the rosette area, primary root length and fresh weight of Cd-stressed *drb2* (*drb2*/Cd) seedlings respectively, compared to *drb2*/Ns seedlings (Figure 1B–D). In addition, the rosette area, primary root length and fresh weight of Cd-stressed *drb4* (*drb4*/Cd) seedlings were decreased by 31.0%, 22.8% and 29.8% respectively, compared to *drb4*/Ns plants (Figure 1B–D). In contrast, the rosette area of Cd-stressed *drb1* (*drb1*/Cd) plants was only mildly reduced by 13.3% compared to the rosette area of *drb1*/Ns seedlings (Figure 1B). Furthermore, and in direct contrast to the negative response of the Col-0, *drb2* and *drb4* primary root to Cd stress, the primary root length of *drb1*/Cd plants was 27.2 mm, compared to 17.8 mm for *drb1*/Ns plants, representing a 53.2% promotion to this assessed parameter (Figure 1C). Furthermore, the mildly reduced rosette area of *drb1*/Cd plants, combined with the increased primary root length, resulted in only a mild reduction (−3.9%) to the biomass of *drb1*/Cd plants compared to *drb1*/Ns plants (Figure 1D). Taken together, the phenotypic assessments presented in Figure 1 clearly show that the *drb1* mutant was the most tolerant to the imposed stress while the *drb2* mutant was the most sensitive of the four assessed *Arabidopsis* lines to Cd stress.

In addition to being reported to repress primary root development, Cd stress has also been demonstrated to inhibit both lateral root formation and elongation [14]. It is therefore important to note here that although not quantified, Figure 1A also clearly shows that lateral root elongation appeared to progress unimpeded in Cd-stressed Col-0, *drb1*, *drb2* and *drb4* seedlings. However, lateral root formation was inhibited by Cd stress in all four assessed plant lines. Figure 1A also depicts that Cd stress induced the formation of adventitious roots to a low degree in the *drb1* mutant and to a high degree in Col-0, *drb2* and *drb4* plants. Although not as striking as the primary root phenotypic difference expressed by the *drb1* mutant compared to those displayed by Col-0, *drb2* and *drb4* plants, this difference in adventitious root formation identified another phenotypic distinction in the response of the *drb1* mutant to the imposed stress.

Total chlorophyll content was next determined for the 15-day-old control and Cd-stressed Col-0, *drb1*, *drb2* and *drb4* seedlings (Figure 1E). Compared to the total chlorophyll content of Col-0/Ns plants of 0.89 milligrams per gram of fresh weight (mg/g FW), total chlorophyll was only mildly reduced in *drb1*/Ns (0.81 mg/g FW) and *drb4*/Ns (0.83 mg/g FW) seedlings and remained at wild-type-equivalent levels in *drb2*/Ns plants (Figure 1E). When compared to the control grown counterpart of each plant line, total chlorophyll content was reduced by 36.0%, 41.6% and 36.1% in Col-0/Cd, *drb2*/Cd and *drb4*/Cd plants, respectively (Figure 1E). In Figure 1A, it can be observed that in contrast to Cd-stressed Col-0, *drb2* and *drb4* seedlings, *drb1*/Cd seedlings maintained their green coloration following the imposed stress. This visual assessment was confirmed via spectrophotometry (Figure 1E), which revealed that the total chlorophyll content of *drb1*/Cd plants was only mildly reduced by 16.6% compared to the total chlorophyll content of *drb1*/Ns plants. 

An Evans blue staining protocol was next applied to determine the degree of cell damage (in the form of membrane disruption) resulting from the production of reactive oxygen species (ROS) in Cd-stressed Col-0, *drb1*, *drb2* and *drb4* seedlings (Figure 1F). Compared to Col-0/Ns plants, the degree of Evans blue staining was elevated by 23.6% in *drb1*/Ns plants, remained unchanged in *drb2*/Ns seedlings and was mildly reduced by 9.5% in the *drb4*/Ns sample (Figure 1F). In the Col-0/Cd sample, compared to the Col-0/Ns sample, the intensity of Evans blue staining increased by 46.6%. Similarly, the intensity of Evans blue staining was increased by 53.2% in the *drb2*/Ns sample compared to the *drb2*/Ns sample. In contrast, the intensity of Evans blue staining only mildly increased by 18.3% to 141.9% in *drb1*/Cd plants from its level of 123.6% in *drb1*/Ns plants (Figure 1F). In the *drb4*/Cd sample, the intensity of Evans blue staining dramatically increased to 264.2% from 90.5% in the *drb4*/Ns sample, to represent a 173.7% increase (Figure 1F). When taken together, the results in Figure 1 clearly demonstrate that the *drb1* mutant was the least sensitive of the four *Arabidopsis* lines assessed in this study to the imposed 7-day Cd stress treatment regime, whereas the *drb2* mutant was the most sensitive of the four *Arabidopsis* plant lines to Cd stress.

### 2.2. Molecular Profiling of the Expression of Cadmium Stress Response Marker Genes

Although a non-essential heavy metal, Cd shares physicochemical properties with the essential heavy metals Cu, Fe and Zn, and as a result, Cd can enter the roots for subsequent transport to the shoots of plants [5,9,10,14]. Evidence of the involvement of a range of heavy metal-responsive proteins in this process stems from the altered expression of their encoding genes upon the plant being exposed to Cd stress [2,3,7,10]. Therefore, a standard reverse transcriptase quantitative polymerase chain reaction (RT-qPCR) approach was used to document any change in the expression of a select group of heavy metal-responsive genes in order to demonstrate that each plant line was experiencing a degree of stress at the molecular level. The Squamosa promoter binding protein-like 7 (SPL7; AT5G18830) transcription factor functions as a central transcriptional regulator when *Arabidopsis* is cultivated in a Cu-deplete environment, and due to SPL7 putatively playing a similar role in Cd stress [5,14], RT-qPCR was used to quantify the abundance of the *SPL7* transcript in control and Cd-stressed Col-0, *drb1*, *drb2* and *drb4* plants. *SPL7* expression was revealed to be moderately reduced by 1.4-, 1.3- and 1.5-fold in *drb1*/Ns, *drb2*/Ns and *drb4*/Ns plants respectively, compared to its expression level in Col-0/Ns plants (Figure 2A). The expression of the *SPL7* locus was also reduced, albeit to a lesser degree (down by 1.1-fold), in Col-0/Cd and *drb1*/Cd seedlings compared to Col-0/Ns and *drb1*/Ns seedlings, respectively. In *drb2*/Cd and *drb4*/Cd plants however, *SPL7* expression was mildly elevated by 1.1-fold (Figure 2A). Altered *SPL7* transcript abundance in the *drb1*, *drb2* and *drb4* control samples and in the Cd-stressed Col-0, *drb1*, *drb2* and *drb4* samples suggested that (1) *SPL7* is indeed responsive to elevated Cd abundance in the growth environment and (2) DRB1, DRB2 and DRB4 potentially play an indirect role in modulating the level of the *SPL7* transcript in *Arabidopsis*.

*Arabidopsis* plants molecularly modified to overexpress the coding sequence of the ATP-binding cassette protein, PDR8 (AT1G59870), are more resistant to Cd stress than unmodified *Arabidopsis* plants [4,15,16], a finding which identifies a role for PDR8 in the response of *Arabidopsis* to Cd stress. RT-qPCR revealed *PDR8* transcript abundance to accumulate to the same level in Col-0/Ns and *drb1*/Ns plants (Figure 2B). In *drb2*/Ns and *drb4*/Ns plants however, *PDR8* expression was reduced by 2.3- and 1.9-fold, respectively. In direct contrast to unchanged and reduced *PDR8* expression in *drb1*/Ns, *drb2*/Ns and *drb4*/Ns plants, *PDR8* expression was elevated by 2.1-, 1.4-, 2.5- and 1.8-fold in Cd-stressed Col-0, *drb1*, *drb2* and *drb4* plants respectively (Figure 2B), compared to the control grown counterpart of each plant line. In species such as *Arabidopsis*, canola and rice, NRAMP proteins have been assigned roles in Cd uptake, transport and/or homeostasis maintenance [2,11,28]. In canola, NRAMP1 is the most responsive *NRAMP* gene family member to Cd stress, with Meng et al. [28] going on to show that in *Arabidopsis*, NRAMP6 is the closest homolog to the canola NRAMP1. RT-qPCR revealed that in the *drb1*/Ns and *drb2*/Ns samples, the abundance of the *NRAMP6* transcript was only mildly elevated by 1.1-fold, whereas in the *drb4*/Ns sample, *NRAMP6* expression was reduced by 1.5-fold (Figure 2C). Compared to the control grown counterpart of each plant line, RT-qPCR revealed *NRAMP6* transcript abundance to only be mildly reduced by 1.1-fold in Col-0/Cd and *drb2*/Cd plants, mildly elevated by 1.2-fold in the *drb1*/Cd sample, and to remain unchanged in *drb4*/Cd plants (Figure 2C). In *Arabidopsis* and canola, Cd stress also alters the expression of the *FE superoxide dismutase 1* (*FSD1*) locus [5,9,14]. Therefore, RT-qPCR was next employed to profile *FSD1* gene expression in control and Cd-stressed Col-0, *drb1*, *drb2* and *drb4* plants, with this approach showing that *FSD1* expression was elevated by 1.3- and 1.2-fold in *drb1*/Ns and *drb2*/Ns plants, respectively (Figure 2D). In contrast, *FSD1* expression was reduced by 1.2-fold in *drb4*/Ns plants. *FSD1* expression was subsequently shown by RT-qPCR to be induced in all assessed lines following Cd stress exposure. More specifically, the expression of *FSD1* was elevated by 1.9-fold in the Col-0/Cd and *drb1*/Cd samples and by 1.3-fold in the *drb2*/Cd and *drb4*/Cd samples (Figure 2D); a finding which clearly demonstrated that the transcriptional activity of the *Arabidopsis FSD1* locus is induced by Cd stress.

*Arabidopsis* encodes eight heavy metal ATPase (HMA) proteins, with the family members HMA2 and HMA4 demonstrated to preferentially efflux Zn ions out of *Arabidopsis* cells [10,29,30]. In addition, HMA2 and HMA4 have both been shown to be able to perform a similar function to efflux Cd [10,29,30], a demonstration that identified these two HMA proteins as ideal candidates to transcriptionally profile via RT-qPCR. Compared to Col-0/Ns plants, *HMA2* expression was decreased by 1.5-fold in *drb1*/Ns plants (Figure 2E). In *drb2*/Ns and *drb4*/Ns plants however, *HMA2* expression increased by 1.8- and 1.2-fold, respectively. In *drb4*/Cd plants, *HMA2* transcript abundance remained unchanged from its already mildly elevated levels in *drb4*/Ns plants. In Col-0/Cd, *drb1*/Cd and *drb2*/Cd plants however, RT-qPCR revealed *HMA2* expression to be elevated by 1.4-, 2.1- and 1.1-fold, respectively (Figure 2E). RT-qPCR next revealed the level of the *HMA4* transcript to be elevated by 1.1- to 1.3-fold in the three control grown *drb* mutant backgrounds compared to its expression level in Col-0/Ns plants (Figure 2F). This transcriptional profiling exercise further revealed that the abundance of the *HMA4* transcript was mildly elevated in all four analyzed *Arabidopsis* lines following their exposure to Cd stress (Figure 2F).

### 2.3. Molecular Profiling of the Response of Auxin Pathway MicroRNAs to Cadmium Stress

Cadmium stress exposure has been demonstrated to alter the abundance of the three miRNAs, miR160, miR167 and miR393, central to the posttranscriptional regulation of the abundance of key pieces of protein machinery of the auxin pathway in canola, maize, radish, rice and sunflower [2,6,7,18,31,32,33]. Compared to Col-0/Ns plants, miR160 abundance was reduced by 6.0-fold in *drb1*/Ns plants and elevated by 2.4- and 2.0-fold in *drb2*/Ns and *drb4*/Ns seedlings, respectively (Figure 3A). RT-qPCR next revealed that Cd stress increased the level of miR160 by 1.2-fold in Col-0/Cd plants. miR160 abundance was also mildly elevated (1.5-fold) in *drb1*/Cd plants; however, in *drb2*/Cd and *drb4*/Cd seedlings, miR160 levels were reduced by 2.0- and 1.4-fold, respectively (Figure 3A). RT-qPCR was next used to document any change to the expression of the three members of the *auxin response factor* (*ARF*) transcription factor gene family, known to be targets of miR160-directed expression regulation, namely *ARF10*, *ARF16* and *ARF17* [34,35]. Compared to Col-0/Ns seedlings, *ARF10* expression was elevated by 2.1-fold in *drb1*/Ns seedlings (Figure 3B) in response to the 6.0-fold reduction in miR160 abundance (Figure 3A). A similar inverse abundance trend for miR160 and *ARF10* was observed in *drb2*/Ns plants; that is, in response to the 2.4-fold elevation in miR160 abundance in *drb2*/Ns plants, *ARF10* expression was reduced by 2.8-fold. In *drb4*/Ns plants however, *ARF10* expression remained at its approximate wild-type level (Figure 3B) in spite of the documented 2.0-fold upregulation of miR160 abundance (Figure 3A). Compared to the Col-0/Ns sample, the mild 20% increase in miR160 levels was revealed by RT-qPCR to result in a 2.1-fold reduction in *ARF10* expression in the Col-0/Cd sample. In *drb1*/Cd seedlings, RT-qPCR further showed a very mild 10% increase in the level of *ARF10* expression (Figure 3B), even though the abundance of miR160 was determined to be 53% higher in *drb1*/Cd seedlings than in *drb1*/Ns seedlings (Figure 3A). A tighter degree of anticorrelation between miR160 and *ARF10* levels was observed in the *drb2*/Cd and *drb4*/Cd samples. More specifically, *ARF10* transcript abundance was elevated by 3.4-fold in response to the 2.0-fold decrease in miR160 abundance in *drb2*/Cd plants, and in *drb4*/Cd plants, *ARF10* expression increased by 1.4-fold in response to the 1.4-fold reduction to the level of miR160 (Figure 3A,B).

The expression profiles constructed for the *ARF16* and *ARF17* miR160 target genes across the four plant lines and two growth regimes assessed in this study were in stark contrast to that generated for the *ARF10* target gene. Namely, *ARF16* expression only moderately differed to the other plant lines and growth conditions assessed in the *drb1*/Ns and *drb1*/Cd samples, with *ARF16* expression upregulated by 1.8- and 1.9-fold in these two samples, respectively (Figure 3C). Elevated *ARF16* expression in *drb1*/Ns and *drb1*/Cd seedlings could be readily accounted for by the 83% and 74% reduction in miR160 abundance in this mutant background across the two analyzed growth regimes (Figure 3A). Similarly, the greatest degree of altered expression was again observed in the *drb1* mutant background with *ARF17* transcript abundance increased by 3.1- and 2.5-fold in *drb1*/Ns and *drb1*/Cd plants, respectively (Figure 3D), with the reduced abundance of miR160 in these two samples (Figure 3A) readily accounting for the documented elevation in *ARF17* expression.

In *Arabidopsis*, *ARF6* and *ARF8* are well-documented posttranscriptional targets of miR167-directed gene expression regulation [36,37]. Therefore, the molecular response of miR167 (Figure 3E), *ARF6* (Figure 3F) and *ARF8* (Figure 3G) to the imposed stress was next investigated. RT-qPCR revealed that compared to Col-0/Ns plants, miR167 levels were reduced by 4.3-, 1.9- and 1.2-fold in *drb1*/Ns, *drb2*/Ns and *drb4*/Ns plants, respectively (Figure 3E). Compared to the control grown counterpart of each *Arabidopsis* line, RT-qPCR next showed miR167 levels to be reduced by 40% and 14% in Col-0/Cd and *drb2*/Cd seedlings respectively, and to remain unchanged in *drb1*/Cd and *drb4*/Cd seedlings (Figure 3E). In response to the large 4.3-fold reduction in miR167 accumulation in *drb1*/Ns plants, *ARF6* expression was mildly elevated by 1.3-fold (Figure 3F). In contrast, *ARF6* transcript abundance was reduced by 1.8- and 1.3-fold in response to the 1.9- and 1.2-fold reduction in miR167 abundance in *drb2*/Ns and *drb4*/Ns plants, respectively. A similar miR167 and *ARF6* expression profile was constructed for Col-0/Cd plants. Namely, *ARF6* expression decreased 1.7-fold in response to the 1.7-fold reduction in miR167 abundance. Compared to the *drb1*/Ns sample, *ARF6* expression was elevated by 1.3-fold in *drb1*/Cd plants (Figure 3F), a mild expression alteration observed in the absence of any change to the level of miR167 (Figure 3E). In contrast, *ARF6* expression was upregulated by 3.0-fold in *drb2*/Cd plants compared to *drb2*/Ns plants in response to the very mild reduction (−14%) in miR167 levels observed in this sample. In *drb4*/Cd plants, *ARF6* expression was also determined to be elevated (by 2.4-fold). However, elevated *ARF6* expression was observed in *drb4*/Cd seedlings in the absence of any change in the abundance of miR167 compared to its levels in *drb4*/Ns seedlings (Figure 3E). The *ARF8* expression profile constructed by RT-qPCR across the *drb1*/Ns, *drb2*/Ns and *drb4*/Ns samples was highly similar to that obtained for the *ARF6* target gene. More specifically, *ARF8* was upregulated by 1.4-fold in *drb1*/Ns plants and downregulated by 2.0- and 1.6-fold in *drb2*/Ns and *drb4*/Ns plants (Figure 3G). Further similarity in the *ARF6* and *ARF8* expression trends was identified across the Col-0/Cd, *drb1*/Cd, *drb2*/Cd and *drb4*/Cd samples; that is, when compared to their respective controls, *ARF8* expression was reduced by 2.0-fold in the Col-0/Cd sample and elevated by 1.1-, 2.0- and 1.6-fold in the *drb1*/Cd, *drb2*/Cd and *drb4*/Cd samples, respectively (Figure 3G).

In addition to the *ARF*s *ARF6*, *ARF8*, *ARF10*, *ARF16* and *ARF17*, forming known targets of miR160- or miR167-directed expression regulation at the posttranscriptional level, the central auxin pathway component TIR1, a protein crucial for auxin perception and signaling, is also under miRNA-directed expression regulation by miR393 [38,39]. RT-qPCR revealed that compared to Col-0/Ns plants, miR393 abundance was decreased by 1.3-fold in *drb1*/Ns plants but was highly increased in its abundance by 3.9- and 4.8-fold in *drb2*/Ns and *drb4*/Ns plants, respectively (Figure 3H). The 7-day Cd stress treatment period mildly lowered the miR393 level by 18.2% in Col-0/Cd plants. The applied stress was also determined to reduce miR393 abundance by 31.9% in the *drb2* mutant. However, compared to the *drb1*/Ns and *drb4*/Ns samples, application of this stress treatment regime enhanced the level of the miR393 sRNA by 31.0% and 25.0% in *drb1*/Cd and *drb4*/Cd plants, respectively (Figure 3H). The expression of *TIR1* was mildly increased by 1.2-fold in response to the 1.3-fold reduction in the level of miR393 in *drb1*/Ns plants. In the *drb2*/Ns sample, *TIR1* expression remained largely unchanged in spite of the significant 3.9-fold elevation in miR393 abundance. Similarly, in spite of the highly enriched accumulation of miR393 (up by 4.8-fold), *TIR1* expression was only mildly reduced by 1.2-fold in *drb4*/Ns plants (Figure 3H,I). Figure 3I also shows that when compared to Col-0/Ns plants, *TIR1* transcript abundance mildly increased by 1.1-fold in Col-0/Cd plants (Figure 3I) in response to the mildly reduced level of miR393 (down by 18.2%) (Figure 3H). Similar transcript abundance trends were determined for miR393 and *TIR1* in *drb2*/Cd plants. Namely, *TIR1* expression was upregulated by 1.9-fold in response to the moderate 32% reduction in miR393 levels. A mild degree of upregulated *TIR1* expression, 1.3-fold, was next documented by RT-qPCR for the *drb1*/Cd sample. However, unlike the Col-0/Cd and *drb2*/Cd samples, elevated *TIR1* expression in *drb1*/Cd plants was in response to elevated (up by 1.3-fold), and not reduced, miR393 abundance (Figure 3H,I). A similar transcript abundance profile was observed for the miR393/*TIR1* expression module in *drb4*/Cd plants compared to *drb4*/Ns plants, with *TIR1* expression upregulated by 1.9-fold (Figure 3I) in response to the 1.3-fold increase in the abundance of miR393 (Figure 3H).

### 2.4. Molecular Profiling of the Response of a Set of Environmental Stress Responsive MicroRNAs to Cadmium Stress

The miRNAs miR395, miR399 and miR408 have been demonstrated to be central to the adaptive response of *Arabidopsis* to the environmental challenges of reduced or eliminated S, PO_4_ and Cu, respectively [40,41,42,43,44,45,46,47]. In addition to these three specific stresses, miR395, miR399 and miR408 have additionally been observed to have altered abundance in canola, maize, radish, rice, soybean and wheat following the exposure of these six plant species to a range of environmental challenges, including elevated Cd [2,6,7,8,9,12,32,33]. The RT-qPCR approach was therefore next applied to uncover any alterations to the miR395, miR399 or miR408 expression modules in control or Cd-stressed Col-0, *drb1*, *drb2* and *drb4* seedlings.

The *Arabidopsis* miR395 expression module was one of the first miRNA expression modules identified to be responsive to abiotic stress, specifically limited S, in plants [40,41]. Therefore, the miR395 expression module was next profiled by RT-qPCR in control and Cd-stressed Col-0, *drb1*, *drb2* and *drb4* seedlings (Figure 4A,B). In *drb1*/Ns plants, miR395 abundance remained at a level equivalent to that detected in Col-0/Ns plants (Figure 4A). However, miR395 abundance was elevated by 3.1- and 5.4-fold in the *drb2*/Ns and *drb4*/Ns samples, respectively. Compared to its abundance in Col-0/Ns plants, the level of miR395 was next revealed by RT-qPCR to be reduced by 1.6-fold in Col-0/Cd seedlings (Figure 4A). Similarly, the 7-day Cd stress treatment regime reduced miR395 abundance by 2.7- and 1.5-fold in *drb1*/Cd and *drb2*/Cd plants, respectively. In contrast however, the level of miR395 remained unchanged in *drb4*/Cd plants (Figure 4A). Of the *ATP sulfurylase* (*ATPS*) genes targeted by miR395 for expression regulation, *ATPS1* (*AT3G22890*) appears to be the most tightly regulated target gene of miR395 in *Arabidopsis*. In *drb1*/Ns plants however, *ATPS1* transcript abundance was determined to be mildly reduced by 1.2-fold, even though miR395 was determined to remain at wild-type approximate levels (Figure 4A,B). In *drb2*/Ns plants, *ATPS1* expression remained unchanged (Figure 4B), in spite of RT-qPCR showing that miR395 abundance was upregulated by 3.1-fold (Figure 4A). In *drb4*/Ns seedlings, *ATPS1* expression was revealed by RT-qPCR to be moderately reduced by 1.6-fold (Figure 4B) in response to the significant 5.4-fold enhancement to miR395 abundance (Figure 4A). In Col-0/Cd seedlings, *ATPS1* transcript levels were reduced by 1.4-fold, a level comparable to the 1.6-fold reduction in miR395 abundance (Figure 4A,B). In the *drb1*/Cd sample, RT-qPCR showed that *ATPS1* expression was elevated by 1.4-fold in response to the 2.7-fold reduction in the level of miR395. Similar to the documented abundance trends reported for miR395 and its target transcript *ATPS1* in Col-0/Cd plants, *ATPS1* expression was decreased by a similar degree, down by 1.6-fold, to the level of reduction observed for miR395 (1.5-fold down) in Cd-stressed *drb2* seedlings. RT-qPCR profiling next revealed that in the *drb4* mutant background, Cd stress failed to further alter the abundance of either miR395 or *ATPS1* compared to their respective levels in *drb4*/Ns seedlings (Figure 4A,B).

The miR399 sRNA is central to the adaptive response of *Arabidopsis* [42,43,44] and of other plant species [45,46] to PO_4_ starvation via its post-transcriptional regulation of the expression of *Phosphate2* (*PHO2*), a gene that encodes a ubiquitin-conjugating enzyme 24 (UBC24). In *drb1*/Ns, *drb2*/Ns and *drb4*/Ns plants, when compared to the Col-0/Ns sample, the level of miR399 was downregulated by 1.3-fold and upregulated by 3.5- and 5.8-fold, respectively (Figure 4C). RT-qPCR further revealed that following the 7-day 50 μM CdCl_2_ stress treatment, miR399 abundance was reduced by 1.7-, 1.3-, 2.9 and 1.2-fold in Col-0/Cd, *drb1*/Cd, *drb2*/Cd and *drb4*/Cd plants, respectively (Figure 4C). In response to the mildly reduced abundance of the regulating miRNA, miR399 (down by 1.3-fold), RT-qPCR revealed *PHO2* expression to be slightly elevated by 1.1-fold in *drb1*/Ns plants (Figure 4D). A similar opposing abundance profile was obtained for miR399 and its *PHO2* target transcript in *drb2*/Ns and *drb4*/Ns plants. More specifically, *PHO2* expression was reduced by 4.3- and 2.0-fold in response to the respective 3.5- and 5.8-fold elevation in miR399 abundance (Figure 4D). In Col-0/Cd plants, *PHO2* transcript abundance was reduced by 2.4-fold (Figure 4D) in response to decreased miR399 abundance (down by 1.7-fold). In contrast to the Col-0/Cd sample, *PHO2* expression was mildly elevated by 1.3-, 1.4- and 1.5-fold in the *drb1*/Cd, *drb2*/Cd and *drb4*/Cd samples, respectively (Figure 4D). Elevated *PHO2* target gene expression in Cd-stressed *drb1*, *drb2* and *drb4* seedlings was in response to the respective 1.3-, 2.9- and 1.2-fold reductions in miR399 abundance (Figure 4C).

In *Arabidopsis*, miR408 is one of three miRNAs (in addition to miR397 and miR857) known to target *Laccase* (*LAC*) genes for expression regulation [47,48], and furthermore, miR408 levels have been shown to be altered when *Arabidopsis* is cultivated in a Cu-deficient environment. Therefore, due to the similar physicochemical properties of Cu and Cd ions, the abundance of miR408 and the expression of its *LAC3* target gene were assessed by RT-qPCR. The accumulation profile constructed for miR408 across the *drb1*/Ns, *drb2*/Ns and *drb4*/Ns samples was highly similar to that obtained for miR399. Namely, miR408 abundance was mildly reduced by 1.1-fold in *drb1*/Ns plants and highly elevated by 3.9- and 6.9-fold in *drb2*/Ns and *drb4*/Ns plants, respectively (Figure 4E). RT-qPCR next showed that the level of miR408 was reduced by 1.7-, 1.4-, 1.9- and 1.3-fold in Col-0, *drb1*, *drb2* and *drb4* plants following their exposure to Cd stress. In *drb1*/Ns and *drb2*/Ns plants, *LAC3* target gene expression was repressed to the same degree, down by 1.3-fold (Figure 4F), even though miR408 abundance was mildly reduced by 1.1-fold in the *drb1*/Ns sample and highly elevated by 3.9-fold in the *drb2*/Ns sample (Figure 4E). The expression of *LAC3* was also mildly elevated by 1.2-fold in *drb4*/Ns seedlings in response to the highly elevated abundance (up by 6.9-fold) of miR408 in this sample (Figure 4E). In Col-0/Cd seedlings, *LAC3* expression was repressed to a greater degree (3.0-fold down) than the targeting miRNA, miR408 (1.7-fold down). In response to the 1.4-fold reduction in miR408 levels in *drb1*/Cd plants (Figure 4E), RT-qPCR revealed *LAC3* transcript abundance to be elevated by 1.6-fold (Figure 4F). The abundance of miR408 was also reduced in *drb2*/Cd and *drb4*/Cd seedlings (Figure 4E); however, *LAC3* transcript abundance was reduced by 1.9- and 4.8-fold respectively by the imposed stress, and not elevated as observed in the *drb1*/Cd sample (Figure 4F).

### 2.5. Molecular Profiling of the Response of the miR396 Expression Module to Cadmium Stress

The Growth regulating factor (GRF) transcription factors form a small family of plant-specific transcription factors, and in *Arabidopsis*, family members *GRF1*, *GRF2*, *GRF3*, *GRF7*, *GRF8* and *GRF9* are known targets of miR396-directed expression regulation [49,50]. Furthermore, in canola, maize, radish and soybean, miR396 has been demonstrated to be responsive to Cd stress [1,6,7,8,33]. Therefore, the entire miR396/*GRF* expression module was profiled by RT-qPCR. In the *drb1*/Ns and *drb4*/Ns samples, miR396 abundance was reduced by 1.9-fold compared to miR396 abundance in the Col-0/Ns sample. In *drb2*/Ns seedlings, miR396 levels were reduced by a greater degree, down by 3.1-fold (Figure 5A). RT-qPCR next revealed that Cd stress reduced miR396 abundance by 2.0-, 1.8-, 2.7- and 2.1-fold in Col-0, *drb1*, *drb2* and *drb4* seedlings, respectively. In response to reduced miR396 abundance in the three assessed *drb* mutants, *GRF1* expression was elevated by 1.9-, 1.5- and 1.4-fold in *drb1*/Ns, *drb2*/Ns and *drb4*/Ns plants, respectively (Figure 5B). In Col-0/Cd and *drb2*/Cd plants, the level of the *GRF1* transcript was reduced by 1.3- and 1.1-fold in response to the 2.0- and 2.7-fold reduction in miR396 abundance. In contrast, *GRF1* expression was enhanced in response to reduced miR396 accumulation in *drb1*/Cd and *drb4*/Cd plants (Figure 5A,B). As determined for *GRF1*, *GRF2* expression was mildly upregulated in the *drb1*/Ns, *drb2*/Ns and *drb4*/Ns samples by 1.2-, 1.4- and 1.1-fold respectively (Figure 5C), in response to reduced miR396 levels in these three *drb* mutants (Figure 5A). In Col-0/Cd, *drb1*/Cd and *drb2*/Cd seedlings, *GRF2* transcript abundance was mildly reduced by 1.4-, 1.2- and 1.3-fold respectively, along with the level of the targeting miRNA, miR396. In *drb4*/Cd plants, *GRF2* expression remained unchanged from its level in *drb4*/Ns plants (Figure 5C), even though miR396 abundance was determined by RT-qPCR to be reduced by 2.1-fold (Figure 5A).

Figure 5D shows that in response to reduced miR396 abundance in control grown *drb1*, *drb2* and *drb4* plants, *GRF3* expression was elevated by 1.6-fold in *drb1*/Ns and *drb2*/Ns plants and further elevated by 2.2-fold in the *drb4*/Ns sample. In Col-0/Cd, *drb1*/Cd and *drb2*/Cd plants, *GRF3* expression was mildly upregulated by 1.2-, 1.1- and 1.1-fold, respectively (Figure 5D). RT-qPCR additionally revealed that the abundance of the *GRF3* transcript remained largely unchanged in the *drb4*/Cd sample compared to its expression level in the *drb4*/Ns sample. Considering that the abundance of miR396 was reduced in all four assessed plant lines following their exposure to Cd stress, the RT-qPCR analysis presented in Figure 5D indicates that the transcriptional activity of the *GRF3* locus is not responsive to Cd stress, nor is the abundance of the *GRF3* transcript overly influenced by the level of miR396. In response to decreased miR396 accumulation in *drb1*/Ns, *drb2*/Ns and *drb4*/Ns seedlings (Figure 5A), *GRF7* expression was determined by RT-qPCR to be elevated by 1.4-fold in the *drb1*/Ns sample and to be highly upregulated by 5.3- and 6.7-fold in the *drb2*/Ns and *drb4*/Ns samples, respectively (Figure 5E). In Col-0/Cd seedlings, *GRF7* expression was downregulated by 1.2-fold in response to decreased miR396 abundance to indicate that the expression of both encoding loci is negatively impacted by elevated Cd in the growth environment (Figure 5A,E). Reduced miR396 abundance in *drb1*/Cd plants, compared to *drb1*/Ns plants, resulted in a 1.8-fold enhancement to the expression level of the *GRF7* gene (Figure 5E). In contrast to the *drb1*/Cd sample, *GRF7* transcript abundance was decreased by 2.5- and 2.4-fold in the *drb2*/Cd and *drb4*/Cd samples respectively (Figure 5E), even though miR396 abundance was also reduced in these two *drb* mutant backgrounds following their exposure to Cd stress. Reduced miR396 and *GRF7* transcript abundance in Cd-stressed Col-0, *drb2* and *drb4* seedlings again indicated that the transcriptional activity of both the *MIR396A*/*B* and *GRF7* loci is repressed by elevated levels of Cd.

RT-qPCR next revealed *GRF8* expression to be upregulated by 2.0-fold, downregulated by 1.4-fold, and to remain unchanged in *drb1*/Ns, *drb2*/Ns and *drb4*/Ns plants, respectively (Figure 5F). Following the 7-day 50 μM CdCl_2_ stress treatment period, in response to reduced miR396 abundance (Figure 5A), *GRF8* expression was mildly elevated by 1.2-fold in Col-0/Cd and *drb1*/Cd seedlings and by 1.3-fold in *drb2*/Cd seedlings (Figure 5F). In response to the 2.1-fold reduction in miR396 abundance in *drb4*/Cd plants, *GRF8* expression was mildly reduced by 1.1-fold (Figure 5F). The abundance of the *GRF9* transcript was revealed by RT-qPCR to be elevated by 1.3-, 1.5- and 1.7-fold in *drb1*/Ns, *drb2*/Ns and *drb4*/Ns plants respectively, in response to the decreased miR396 accumulation documented for these three *drb* mutants (Figure 5G). When compared to the control grown counterpart of each plant line profiled via RT-qPCR, *GRF9* target gene expression remained unchanged in Col-0/Cd and *drb2*/Cd plants and was mildly reduced and elevated by 1.1- and 1.3-fold respectively, in *drb1*/Cd and *drb4*/Cd plants (Figure 5G). Considering that the abundance of the regulating miRNA miR396 was reduced in Col-0, *drb1*, *drb2* and *drb4* seedlings following their exposure to Cd stress (Figure 5A), the unchanged-to-mild alteration to *GRF8* (Figure 5F) and *GRF9* (Figure 5G) transcript abundance in the four assessed *Arabidopsis* lines indicated that miR396 is not a potent post-transcriptional regulator of *GRF8* or *GRF9* expression following the application of Cd stress.

## 3. Discussion

Cadmium is a natural constituent of the earth’s crust, and as such, Cd is routinely released into the environment due to rock weathering, soil erosion and volcanic events [4,5]. However, in recent decades, the rate of release of Cd into the environment has increased greatly due to anthropogenic activities, namely agricultural and industrial practices [6,7,8]. Cd is not an essential heavy metal; however, due to the shared physicochemical properties of Cd ions to those of the essential heavy metals Cu, Fe and Zn, Cd is readily taken up by the root system of a plant for its subsequent translocation to the aerial tissues [2,7,9,10]. As a non-essential heavy metal, Cd accumulation in the edible parts of a plant is highly detrimental to human and animal health, with the continued accumulation of Cd in human tissues posing serious health concerns [7,10,11,12]. In an attempt to equilibrate, and/or sequester away Cd ions to neutralize their toxic effects, plants rely on a highly complex and interrelated molecular network [1,2,4,7,9,11,12,13,14]. At the post-transcriptional level, the miRNA class of small regulatory RNA has been identified as a central regulator of the gene expression changes required for a plant to attempt to mount an adaptive response to Cd stress, a regulatory role that has been demonstrated across a wide range of evolutionary distantly related plant species [2,4,6,7,8,9,12,14,15,18]. Therefore, in this study, we assessed the role of miRNA-directed alterations to target gene expression in response to a 7-day 50 μM CdCl_2_ treatment regime of 8-day old Col-0, *drb1*, *drb2* and *drb4* seedlings to determine the degree of involvement of this mode of posttranscriptional gene expression regulation in the response of *Arabidopsis* to growth in the presence of Cd. The three *drb* mutant lines *drb1*, *drb2* and *drb4* were included in the analyses reported here due the central requirement of DRB1, DRB2 and DRB4 in miRNA production and/or miRNA action (i.e., target gene expression regulation) in *Arabidopsis* [19,20,21,22,23,24,25,26,27].

### 3.1. The Arabidopsis Plant Lines Defective in the Activity of MicroRNA Pathway Machinery Proteins Display Differing Degrees of Sensitivity to Cadmium Stress at Both the Phenotypic and Molecular Levels

Cultivation of 8-day-old Col-0, *drb1*, *drb2* and *drb4* seedlings for a 7-day period in the presence of 50 μM CdCl_2_ clearly revealed that of the four *Arabidopsis* plant lines assessed, the *drb1* mutant was the least sensitive to the imposed stress (Figure 1A) as evidenced by the mildest reduction in the rosette area (Figure 1B), fresh weight (Figure 1D) and total chlorophyll (Figure 1F) metrics in Cd-stressed *drb1* plants. The most visually striking distinction of Cd-stressed *drb1* plants from Cd-stressed Col-0, *drb2* and *drb4* seedlings was the promotion of primary root development (Figure 1A,C); a phenotype that is in complete contrast to the previously reported phenotypic consequence of inhibition of primary root elongation in Cd-stressed *Arabidopsis* and wheat seedlings [12,14]. A second clear phenotypic distinction displayed by the root system of Cd-stressed *drb1* seedlings was the failure of *drb1*/Cd plants to form adventitious roots, with adventitious root formation clearly induced in Cd-stressed Col-0, *drb2* and *drb4* plants (Figure 1A). In addition, the phenotypic and physiological analyses presented in Figure 1 also show that the *drb4* mutant displayed a similar degree of sensitivity to Cd stress as did wild-type *Arabidopsis* (Col-0) seedlings, and that the *drb2* mutant line was the most sensitive to the imposed stress.

The range of phenotypic and physiological responses displayed by Col-0/Cd, *drb1*/Cd, *drb2*/Cd and *drb4*/Cd seedlings strongly indicates that each plant line was indeed stressed, albeit to different degrees, by its cultivation for a 7-day period in the presence of 50 μM CdCl_2_. To next demonstrate that the four assessed *Arabidopsis* lines were additionally responding at the molecular level to the imposed stress, the expression of six loci known to be responsive to heavy metals [1,2,4,5,7,9,15], including *SPL7*, *PDR8*, *NRAMP6*, *FSD1*, *HMA2* and *HMA4*, was next profiled via RT-qPCR (Figure 2). This analysis revealed that indeed, Col-0, *drb1*, *drb2* and *drb4* seedlings were responsive at the molecular level to Cd stress, as evidenced by the mild expression changes observed for *SPL7* (Figure 2A), *NRAMP6* (Figure 2C) and *HMA4* (Figure 2F), together with the moderate degree of expression alteration documented for *PRD8* (Figure 2B), *FSD1* (Figure 2D) and *HMA2* (Figure 2E). Previous genetic-based studies in *Arabidopsis* where the expression of *NRAMP6*, *PRD8* or *SPL7* was molecularly manipulated to be either increased (*PRD8*) [4,15,16] or repressed (*NRAMP6* and *SPL7*) [4,5,14,15,17] have provided strong evidence of the involvement of a range of proteins required for the uptake, translocation and/or sequestration of essential heavy metals such as Cu and Fe in the response of *Arabidopsis* to the non-essential heavy metal Cd. Therefore, the expression trends presented in Figure 2 for *SPL7*, *PDR8*, *NRAMP6*, *FSD1*, *HMA2* and *HMA4*, (1) further identify a putative, similar functional role for the encoded proteins of these heavy metal responsive genes in the response of *Arabidopsis* to Cd stress, as well as to (2) readily demonstrate that a molecular response was indeed elicited by the imposed stress in all four assessed *Arabidopsis* lines.

### 3.2. DRB1 Is the Primary DRB Protein Required for the Production of MicroRNAs Responsive to Cadmium Stress

The *drb1* mutant line is defective in the activity of DRB1, the preferred functional partner protein of DCL1 for the production of the majority of *Arabidopsis* miRNAs [19,21,22]. DRB1 ensures that DCL1 is correctly positioned on the miRNA precursor transcript for accurate DCL1-catalyzed miRNA production [19,21,22]. Therefore, in the absence of DRB1, processing accuracy is lost, resulting in reduced mature miRNA accumulation in the *drb1* mutant background [19,21,22]. RT-qPCR quantification of miRNA abundance in control grown Col-0, *drb1*, *drb2* and *drb4* plants clearly revealed that five of the seven miRNAs assessed, including miR160, miR167, miR393, miR399 and miR408, accumulated to their lowest level in *drb1*/Ns plants, a finding that identified DRB1 as the primary DRB protein required for the production of these five miRNAs. The S-responsive miRNA, miR395, was however revealed to remain at levels equivalent to those documented for Col-0/Ns plants in *drb1*/Ns seedlings (Figure 4A). In spite of this observation, the high degree of miR395 abundance upregulation observed in *drb2*/Ns and *drb4*/Ns plants, suspected to be the result of the antagonistic action of the DRB2 and DRB4 proteins on DRB1 function, did however indicate that DRB1 is also the primary DRB protein required for miR395 production in *Arabidopsis*.

In contrast to our finding that DRB1 is the primary DRB protein required for miR160, miR167, miR393, miR395, miR399 and miR408 production, the seventh miRNA quantified in this study, miR396, accumulated to its lowest abundance level in the *drb2*/Ns sample (Figure 5A). Although miR396 accumulation was also reduced in the *drb1*/Ns and *drb4*/Ns samples, the higher degree of reduction in miR396 abundance in the *drb2*/Ns sample identified DRB2 as the primary DRB protein required for the production of this miRNA in *Arabidopsis*. We have previously demonstrated [26] that DRB2 is capable of repressing *DRB1* gene expression, an expression alteration that would reduce the abundance of the DRB1 protein, the preferred functional partner of DCL1 [20,21]. Reduced DRB1 abundance would, in turn, allow DRB2 to outcompete the residual DRB1 protein that remained for its functional interaction with DCL1, and with select miRNA precursor transcripts, for the production of a specific subset of *Arabidopsis* miRNAs. Therefore, the identification that DRB2 is the primary DRB protein required for miR396 production in *Arabidopsis* forms a highly interesting result which warrants further functional characterization in the future.

Quantification of miRNA abundance via RT-qPCR additionally identified secondary roles for both DRB2 and/or DRB4 in the production of miR160, miR167, miR393, miR395, miR399 and miR408. The secondary requirement to DRB1 function for DRB2 and DRB4 in the production of the miRNAs assessed in this study was provided via the altered accumulation of six of the seven miRNAs analyzed in the *drb2*/Ns and *drb4*/Ns samples. Enhanced miRNA abundance in the *drb2* and *drb4* mutants stems from the removal of the DRB2 and/or DRB4 antagonistic action on DRB1 function in the DRB1/DCL1 partnership, a complex interrelationship that has previously been reported for the production of other *Arabidopsis* miRNAs [24,25,26,27,44] and that is thought to be required to ensure that the level of each mature miRNA is exactly correct throughout *Arabidopsis* development. Further evidence of the complexity of miRNA production in *Arabidopsis* is provided by the accumulation profile generated for miR167 across control grown *drb1*, *drb2* and *drb4* seedlings. Namely, miR167 accumulation was reduced by 4.4-, 1.9- and 1.2-fold in *drb1*/Ns, *drb2*/Ns and *drb4*/Ns seedlings, respectively (Figure 3E), revealing that DRB1 is the primary DRB protein required for miR167 production in *Arabidopsis*, but that DRB2 and DRB4 also contribute to this process to ensure that miR167 accumulates to its correct level. Similarly, although DRB2 was identified to be the primary DRB protein required for miR396 production, the 1.9-fold reduction in miR396 abundance detected in both the *drb1*/Ns and *drb4*/Ns samples suggested that in wild-type *Arabidopsis*, DRB1 and DRB4 mediate secondary roles to DRB2 in order to “fine-tune” miR396 production.

### 3.3. RT-qPCR Profiling of Altered MicroRNA Abundance in Response to Cadmium Stress

In addition to being applied to uncover the requirement of DRB1, DRB2 and/or DRB4 function for miRNA production, RT-qPCR was used to document alterations to the abundance of miR160, miR167, miR393, miR395, miR396, miR399 and miR408 in *Arabidopsis* exposed to Cd stress. In sunflower [18] and maize [32], Cd stress has been reported to enhance miR160 accumulation. A matching abundance response was observed here in Cd-stressed Col-0 and *drb1* seedlings (Figure 3A) to show that in *Arabidopsis*, miR160 abundance is elevated in response to the presence of Cd in the growth environment. In contrast to miR160, miR167 abundance was reduced in 15-day-old Col-0 and *drb2* seedlings following the imposed stress (Figure 3E), with reduced miR167 levels previously reported in maize following Cd stress [6,32]. In canola, radish and sunflower however, miR167 abundance showed the opposing trend of elevated accumulation in response to Cd stress [2,11,18]. Repression in *Arabidopsis* and maize, versus elevation in canola, radish and sunflower, strongly suggests that these differential miR167 accumulation trends likely stem from the differing composition of the *cis*-element regulatory landscapes of the promoter regions that control *MIR167* gene expression in these five plant species. Similarly, Cd stress was revealed to mildly reduce miR393 abundance in Col-0 seedlings (Figure 3H), with decreased miR393 accumulation in response to Cd stress demonstrated previously in canola [1,33]. In alfalfa, maize and radish however, miR393 accumulation has been demonstrated to be elevated in response to Cd stress [6,7,18,32]. The differential response of miR393 across these five plant species further identifies the importance of the composition of the *cis*-element landscape within the regulatory sequences, namely the promoter region, of the encoding *MIR393* loci of different plant species to direct a specific transcriptional response to this form of heavy metal stress.

The accumulation of miR395 is well documented to increase in *Arabidopsis* following its cultivation in a growth environment deplete of S [40,41]. Here, we show an opposing abundance trend for miR395 (Figure 4A) following the cultivation of Col-0, *drb1* and *drb2* seedlings for a 7-day period in the presence of 50 μM CdCl_2_. Decreased miR395 accumulation in response to elevated Cd has been reported previously for wild-type *Arabidopsis* plants, as well as for Cd-stressed canola [2] and radish [7]. Similar to the response of miR395 to Cd stress, decreased miR396 accumulation has been reported previously in canola, radish, maize and soybean plants in response to Cd stress [1,6,7,8,33]. Figure 5A shows that the level of miR396 was reduced in all four assessed *Arabidopsis* lines following Cd stress application. When taken together with the matching miR396 accumulation trends previously reported for canola, radish, maize and soybean plants exposed to Cd stress [1,6,7,8,33], reduced miR396 abundance, and therefore, the release of the post-transcriptional regulation of the expression of its *GRF* target genes, identifies the miR396 expression module as playing a crucial role in the response of a plant to Cd stress.

Via comparison to the control grown counterpart of each assessed *Arabidopsis* line, RT-qPCR revealed miR399 abundance to be reduced in response to the 7-day Cd stress treatment regime imposed on 8-day-old Col-0, *drb1*, *drb2* and *drb4* seedlings (Figure 4C). Altered miR399 abundance in response to elevated Cd has been reported previously in canola [9], maize [32] and rice [12], reports which further confirm that miR399 is indeed responsive to elevated Cd in addition to being induced in its abundance by PO_4_ starvation [42,43,44]. Similar to the miR399 abundance trends documented for Cd-stressed Col-0, *drb1*, *drb2* and *drb4* seedlings, miR408 accumulation was revealed by RT-qPCR to be reduced in each of the four *Arabidopsis* lines analyzed in this study (Figure 4E). Altered miR408 abundance in response to the presence of Cd in the growth environment of *Arabidopsis* likely stems from an indirect transcriptional response of the *MIR408* locus due to the physicochemical properties shared between Cd and Cu ions with miR408 accumulation demonstrated to be induced when *Arabidopsis* is cultivated in a growth environment that is deplete in Cu [47,48].

### 3.4. RT-qPCR Profiling of the Expression of Target Genes of Cadmium-Responsive MicroRNAs

Considering our previous demonstration that DRB1 and DRB2 act as molecular switches to determine the mechanism of RNA silencing directed by a specific cohort of *Arabidopsis* miRNAs [24,25,26,27], RT-qPCR was next applied to attempt to gain further insight into the potential mechanism of gene expression regulation directed by each of the seven *Arabidopsis* miRNAs identified here as Cd stress-responsive. For the three miR160 target genes, including *ARF10*, *ARF16* and *ARF17* [34,35], profiled by RT-qPCR in 15-day-old control or Cd-stressed Col-0, *drb1*, *drb2* and *drb4* plants, transcript cleavage was revealed to be the predominant mechanism of RNA silencing directed by miR160 (Figure 3B–D). For example, compared to its abundance in Col-0/Ns seedlings, miR160 levels were reduced and elevated in *drb1*/Ns and *drb2*/Ns seedlings, respectively (Figure 3A). Accordingly, *ARF10* expression was enhanced and repressed in *drb1*/Ns and *drb2*/Ns seedlings respectively (Figure 3B), due to the relaxation and contraction of the stringency of miR160-directed target gene expression regulation. In addition to the miR160 target genes *ARF10*, *ARF16* and *ARF17*, the *ARF6* and *ARF8* transcripts form well-documented targets of miR167-directed gene expression regulation in *Arabidopsis* [36,37]. Comparison of *ARF6* and *ARF8* expression between Col-0/Ns and *drb1*/Ns seedlings suggested that miR167 controls the abundance of both of its *ARF* gene targets via a transcript cleavage mechanism of miRNA-directed gene expression regulation. Namely, in response to the 4.3-fold reduction in miR167 abundance in *drb1*/Ns seedlings compared to its levels in Col-0/Ns seedlings, *ARF6* and *ARF8* expression was mildly elevated by 1.3- and 1.4-fold, respectively (Figure 3E–G). It is important to note here however, that in spite of this clear reciprocal relationship between miR167 and its *ARF6* and *ARF8* target transcripts in Col-0 and *drb1* control seedlings, miR167-directed expression regulation of *ARF6* and *ARF8* appeared to become highly defective in the *drb2* and *drb4* mutant backgrounds and in both assessed growth regimes (Figure 3E–G). In alfalfa, *Arabidopsis*, maize and radish, *TIR1* has been identified as a posttranscriptional target of miR393-directed expression regulation [6,7,18,32,33,38,39]. Again, via comparison of the abundance of the miR393 and *TIR1* transcripts in Col-0/Ns and *drb1*/Ns seedlings, mildly reduced levels of miR393 were revealed to result in mildly elevated *TIR1* expression (Figure 3H,I). This RT-qPCR-generated finding tentatively suggested that target transcript cleavage is the gene expression mechanism directed by miR393 to control the abundance of the *TIR1* mRNA in 15-day-old *Arabidopsis* whole seedlings. In addition, as outlined for the miR167/*ARF6*/*ARF8* expression module, miR393-directed regulation of *TIR1* expression appeared to become defective in the *drb2* and *drb4* mutant backgrounds and in both assessed growth regimes.

In addition to having its accumulation induced by limited S, miR395 abundance has been reported to be negatively altered in *Arabidopsis*, canola and radish in response to elevated Cd [2,7,14,40,41], an accumulation trend confirmed here for Cd-stressed Col-0, *drb1*, *drb2* and *drb4* seedlings (Figure 4A). Reduced miR395 abundance would result in the release of miR395-directed expression regulation of *ATPS1* and of the other *ATPS* transcripts additionally targeted by this miRNA. Loss or partial relaxation of *ATPS* expression regulation when *Arabidopsis* is experiencing Cd stress would allow for an adequate supply of S-containing compounds to continue to be produced in order for *Arabidopsis* to attempt to detoxify the cellular effects induced by the over-accumulation of this non-essential heavy metal. Although miR395 was demonstrated to be responsive to the imposed stress, *ATPS1* expression showed little anticorrelation to the abundance of its targeting miRNA across the four *Arabidopsis* lines and two growth regimes assessed in this study (Figure 4B). This finding suggests that in 15-day-old *Arabidopsis* seedlings, *ATPS1* expression is not overly influenced by the abundance of its targeting sRNA, miR395. In direct contrast to the miR395/*ATPS1* relationship, the mild elevation in *PHO2* transcript abundance in *drb1*/Ns plants, in combination with the significant repression of *PHO2* expression in *drb2*/Ns and *drb4*/Ns seedlings (Figure 4D), revealed tight reciprocal target gene expression trends to the level of miR399 (Figure 4C). The reciprocal abundance trends for miR399 and *PHO2* transcripts revealed that miRNA-directed target transcript cleavage is the predominant mode of *PHO2* expression regulation directed by miR399 in control grown *Arabidopsis*. Furthermore, the abundance of the *PHO2* transcript was elevated in Cd-stressed *drb1*, *drb2* and *drb4* seedlings in response to reduced levels of the targeting miRNA in each mutant background (Figure 4C,D), a finding that provides additional support to the concept that the predominant mode of *PHO2* expression regulation directed by miR399 in *Arabidopsis* is target transcript cleavage. In Cd-stressed Col-0, *drb1*, *drb2* and *drb4* seedlings, both the abundance of the miR408 sRNA and the *LAC3* target transcript were reduced (Figure 4E,F). Reduction in the level of the targeting miRNA and of the *LAC3* target transcript suggested that the transcriptional activity of both the *MIR408* and *LAC3* loci is repressed by the imposed stress.

Figure 5 clearly shows that although DRB2 was identified as the primary DRB protein required for miR396 production (Figure 5A), and not DRB1 as demonstrated for the other six miRNAs analyzed in this study (Figure 3 and Figure 4), transcript cleavage formed the dominant mode of miR396-directed *GRF* target gene expression regulation in the *drb1*, *drb2* and *drb4* mutant backgrounds (Figure 5B–G). In addition to our previous demonstration that DRB2 is required for the production of a specific cohort of *Arabidopsis* miRNAs, we subsequently showed that this specific miRNA cohort regulates the abundance of their target gene transcripts via the alternate translational repression mode of RNA silencing [26,27]. However, the expression analyses presented in Figure 5 clearly demonstrate that this is not the mechanism of RNA silencing directed by miR396 to regulate the expression of its *GRF* target genes. More specifically, comparison of miR396 abundance to *GRF* target gene expression in the control grown counterpart of each analyzed *Arabidopsis* line exposed to the Cd stress treatment regime clearly revealed that miR396 directs transcript cleavage to regulate the expression of its target genes, *GRF1*, *GRF2*, *GRF3*, *GRF7* and *GRF8*, whereas the expression level of *GRF9* appeared not to be overly influenced by miR396 abundance.

## 4. Materials and Methods

### 4.1. Arabidopsis Plant Lines and Cadmium Stress Treatment

The *Arabidopsis* plant lines harboring Transfer-DNA (T-DNA) insertion mutations used in this study, including the *drb1* (*drb1-1*; SALK_064863), *drb2* (*drb2-1*; GABI_348A09) and *drb4* (*drb4-1*; SALK_000736) mutants, have been described previously [21,22,23,24,25]. The *drb* mutant seeds and those of unmodified wild-type *Arabidopsis* (ecotype Columbia-0 (Col-0)) plants were surface-sterilized using chlorine gas. Following sterilization, seeds were plated onto standard solid *Arabidopsis* growth medium (half-strength Murashige and Skoog (MS) salts), and then, the plates containing the growth medium were sealed with gas-permeable tape and incubated in the dark for 48 h (h) at 4 °C for stratification. Post stratification, the sealed plates were transferred to a temperature-controlled growth cabinet (A1000 Growth Chamber, Conviron^®^, Melbourne, Australia) and cultivated for eight days under a standard growth regime of 16 h light/8 h dark, and a 22 °C/18 °C day/night temperature. At day 8, equal numbers (n = 48; 4 × plates of 12 seedlings per plate) of Col-0, *drb1*, *drb2* and *drb4* seedlings were transferred to either (1) a fresh plate of standard *Arabidopsis* plant growth medium (control plants (Ns plants)) or (2) a fresh plate of solid *Arabidopsis* growth medium that had been supplemented with 50 micromolar (μM) cadmium chloride (CdCl_2_) (Cd-stressed plants (Cd plants)). Following seedling transfer, the Ns and Cd representative seedlings of each *Arabidopsis* line were returned to the temperature-controlled growth cabinet for an additional 7-day period. The concentration of CdCl_2_, and the duration of the Cd stress treatment period applied in this study, were selected to provide a prolonged mild-to-moderate phenotypic response in the four *Arabidopsis* lines under assessment based on previously reported experimentation [51,52]. In addition, and in order to prepare the seedlings for the reported phenotypic assessments, it is important to note here that for this 7-day cultivation period, the media plates were orientated for vertical growth.

### 4.2. Phenotypic and Physiological Assessments

The fresh weight of 15-day-old Col-0, *drb1*, *drb2* and *drb4* whole seedlings cultivated on either standard *Arabidopsis* plant growth medium (Ns plants) or Cd stress media (Cd plants) was recorded in order to determine the influence of the presence of Cd in the growth environment on *Arabidopsis* development. In addition, the area of the rosette and the length of the primary root of 15-day-old control and Cd-stressed Col-0, *drb1*, *drb2* and *drb4* plants was determined via the assessment of photographic images using ImageJ software.

For the quantification of the total content of chlorophyll (namely chlorophylls *a* and *b* combined), the rosette leaves of 15-day-old Ns and Cd Col-0, *drb1*, *drb2* and *drb4* plants were sampled and incubated in the dark for 24 h in a solution of 80% (*v*/*v*) acetone. After the 24-h incubation period, samples were centrifuged at room temperature for 7 minutes (min) at 15,000× *g* to clarify the solutions. The resulting supernatants were then transferred to a spectrophotometer (Thermo Scientific, Sydney, Australia) to determine the absorbance of each sample at 646 and 663 nanometer (nm) wavelengths, and using 80% (*v*/*v*) acetone as the blanking solution. Lichtenthaler’s equations were used to convert the initially recorded absorbance values to micrograms per gram of fresh weight (μg/g FW) for total chlorophyll determination exactly as outlined in [53].

The degree of oxidative stress, and therefore cellular damage, induced by the cultivation of Col-0, *drb1*, *drb2* and *drb4* seedlings for a 7-day period in the presence of 50 μM CdCl_2_ was determined using the protocol described in [54]. In brief, 200 mg of plant tissue was sampled from each plant line, and for the two assessed growth regimes, and post-sampling, tissues were crushed on, and incubated on ice for 20 min in 0.25% (*v*/*v*) of Evans blue solution. The samples were subsequently washed twice for 15 min per wash in distilled water. The washed tissue samples were then homogenized further using a chilled mortar and pestle that contained 50 μL of freshly prepared de-staining solution (50:49:1 *v*/*v*/*v* of 100% ethanol/distilled water/10% (*v*/*v*) sodium dodecyl sulfate (SDS)). Samples were incubated at 50 °C for 15 mins and then centrifuged for 15 min at room temperature at 15,000× *g* to pellet the cellular debris. The resulting supernatants were diluted 1:10 in de-staining solution and then immediately transferred to a spectrophotometer (Thermo Scientific, Sydney, Australia) to determine the absorbance of each sample at 600 nm via the use of the de-staining solution as the blanking solution.

### 4.3. Total RNA Extraction for Quantitative Molecular Assessments

For the reported molecular assessments, total RNA was extracted from four biological replicates of pools of six individual plants of 15-day-old control grown and Cd-stressed Col-0, *drb1*, *drb2* and *drb4* plants using TRIzol^TM^ Reagent according to the protocol of the manufacturer (ThermoFisher Scientific, Sydney, Australia). It is important to note here that the 24 plants, per plant line and growth regime, used for total RNA extraction differed to those used for the physiological analyses due to the destructive nature of all experiments performed. The quality of the extracted total RNA was assessed via standard electrophoretic separation of the nucleic acid on an ethidium bromide-stained 1.2% (*w*/*v*) agarose gel. For each high-quality total RNA preparation, a NanoDrop spectrophotometer (NanoDrop^®^ ND-1000, Thermo Scientific, Sydney, Australia) was subsequently employed to determine total RNA concentration in micrograms per microliter (μg/μL).

A global, high molecular weight complementary DNA (cDNA) library for gene expression quantification was constructed via the digestion of 5.0 μg of total RNA with 5.0 units (U) of DNase I according to the manufacturer’s instructions (New England Biolabs, Melbourne, Australia). The DNase I-treated total RNA was next purified using a RNeasy Mini Kit (Qiagen, Melbourne, Australia) according to the manufacturer’s protocol, and 1.0 μg of this purified preparation was then used as the template to synthesize cDNA via the use of 1.0 U of the ProtoScript^®^ II Reverse Transcriptase and 2.5 mM of oligo dT_(18)_ according to the manufacturer’s instructions (New England Biolabs, Melbourne, Australia).

MicroRNA-specific cDNAs were synthesized via the treatment of 500 nanograms (ng) of total RNA with 0.2 U of DNase I (New England Biolabs, Melbourne, Australia). Each DNase I-treated total RNA sample was directly used as a template for miRNA-specific cDNA synthesis using miRNA-specific stem-loop DNA oligonucleotides (Appendix A) and 1.0 U of ProtoScript^®^ II Reverse Transcriptase (New England Biolabs, Melbourne, Australia). The cycling conditions of (1) 1 cycle of 16 °C for 30 min, (2) 60 cycles of 30 °C for 30 s (s), 42 °C for 30 s and 50 °C for 2 s, and (3) 1 cycle of 85 °C for 5 min were used for miRNA-specific cDNA synthesis.

All generated single-stranded cDNAs were subsequently diluted to a working concentration of 50 ng/μL in RNase-free water prior to their use as a template for the quantification of the abundance of either a gene transcript or specific miRNA. In addition, all RT-qPCRs used the same cycling conditions of (1) 1 cycle of 95 °C for 10 min, and (2) 45 cycles of 95 °C for 10 s and 60 °C for 15 s. The GoTaq^®^ qPCR Master Mix (Promega, Sydney, Australia) was used as the fluorescent reagent for all performed RT-qPCR experiments. miRNA abundance and gene transcript expression were quantified using the 2^−ΔΔCT^ method with the small nucleolar RNA, *snoR101*, and *Ubiquitin10* (*UBI10*; *AT4G05320*) used as the respective internal controls to normalize the relative abundance of each assessed transcript. For all RT-qPCR experiments reported here, four biological replicates were used per sample, and three technical replicates were performed per biological replicate. The sequence of each DNA oligonucleotide used in this study either for the synthesis of a miRNA-specific cDNA, or to quantify gene transcript abundance via RT-qPCR is provided in Appendix A.

### 4.4. Statisical Analysis

Analytical data from this study were obtained from at least 4 biological replicates. Statistical analysis was performed using the one-way analysis of variance (ANOVA; RRID:SCR_002427) method while the post-hoc Tukey test was performed using the SPSS Program (IBM, United States; RRID:SCR_002865). The results of these analyses are presented as letters above the columns on the relevant histograms. The same letter indicates a non-statistically significant difference (*p* > 0.05), whereas a different letter indicates a statistically significant difference (*p* < 0.05).

## 5. Conclusions

Here we report on the degree of involvement of miRNA-directed molecular responses to a 7-day 50 μM CdCl_2_ stress treatment regime in 15-day-old seedlings of wild-type *Arabidopsis* (Col-0) plants and the *drb1*, *drb2* and *drb4* single mutants. The phenotypic and physiological assessments of these four *Arabidopsis* lines clearly revealed that the *drb1* mutant was the least sensitive to the imposed stress, while the *drb2* mutant was identified as the most sensitive to Cd stress. Subsequent molecular profiling of Cd-stressed Col-0, *drb1*, *drb2* and *drb4* seedlings, for comparison to their respective control grown counterparts, showed that DRB1 was the primary DRB protein required for the production of six (namely miR160, miR167, miR393, miR395, miR399 and miR408) of the seven miRNAs assessed in this study. However, for all six of these miRNAs, DRB2 and DRB4 were demonstrated to play secondary regulatory roles in their production. The seventh miRNA analyzed in this study, miR396, was revealed to require DRB2, and not DRB1, for its production. However, as demonstrated for the six DRB1-dependent miRNAs analyzed, DRB1 and DRB4 were revealed to mediate secondary roles in regulating miR396 production; an additional tier of regulation suspected to be placed on each assessed miRNA to ensure its correct abundance throughout *Arabidopsis* development, or when *Arabidopsis* is exposed to environmental stress. Although DRB2 was demonstrated to be the primary DRB protein required for miR396 production, in combination with its secondary role in miR160, miR167, miR393, miR395, miR399 and miR408 production, it was somewhat unexpected to subsequently determine that miRNA-directed target transcript cleavage appeared to be the sole mechanism of target gene expression regulation directed by each of the seven assessed miRNAs. In summary, the previous demonstration that these seven miRNAs are responsive to other forms of abiotic and/or biotic stress in *Arabidopsis* and other plant species, together with the functional diversity of the proteins encoded by the target genes of this miRNA cohort, provides additional insight into the complexity of the miRNA-directed molecular response of *Arabidopsis* to Cd stress as a small part in the attempt of the *Arabidopsis* plant to mount an adaptive response to Cd stress.

## Figures and Tables

**Figure 1 plants-10-00130-f001:**
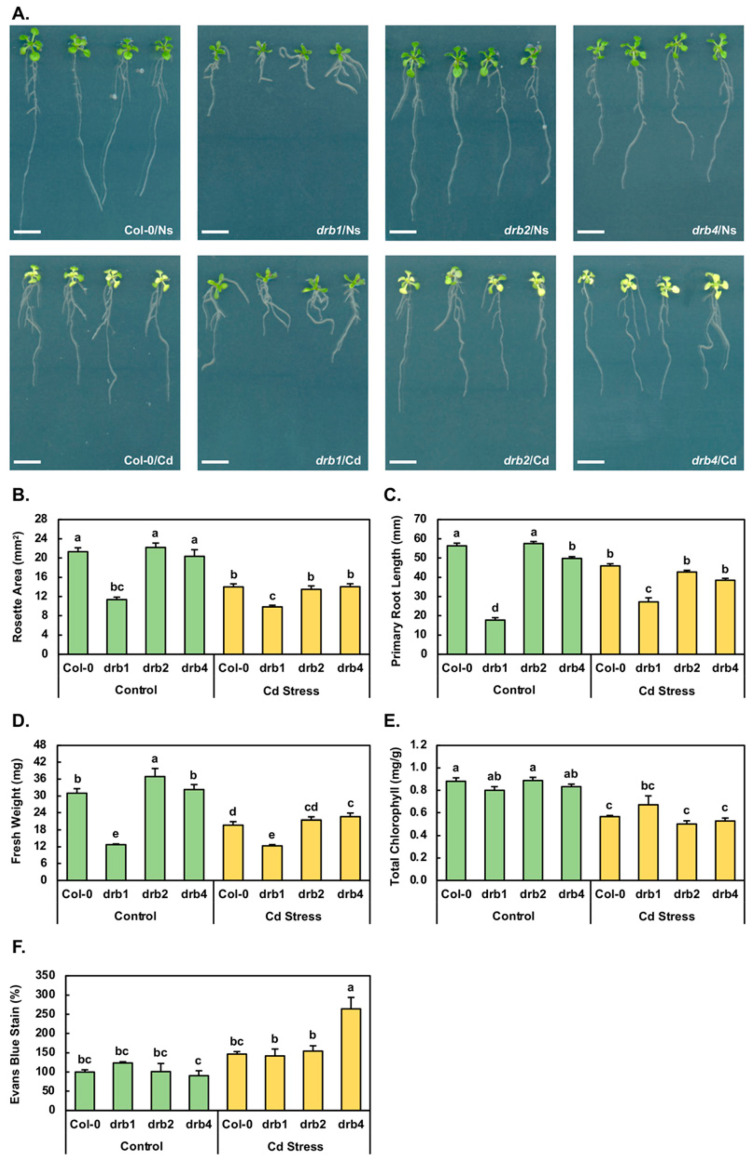
The phenotypic and physiological responses of the *Arabidopsis* lines Col-0, *drb1*, *drb2* and *drb4* to cadmium stress. (**A**) The phenotypes displayed by 15-day-old control (top panels) and Cd-stressed (bottom panels) Col-0, *drb1*, *drb2* and *drb4* plants. Bar = 1.0 cm. Quantification of the phenotypic parameters, (**B**) rosette area (mm^2^), (**C**) primary root length (mm) and (**D**) whole seedling fresh weight (mg), of 15-day-old Cd-stressed Col-0, *drb1*, *drb2* and *drb4* seedlings compared to those of the control grown counterpart of each plant line. Determination of the physiological parameters, (**E**) total chlorophyll content (mg/g FW) and the (**F**) degree of cell membrane damage (presented as a percentage (%) of the absorbance of the Col-0/Ns sample at wavelength 600 nm), as determined by spectrophotometry of control and Cd-stressed Col-0, *drb1*, *drb2* and *drb4* seedlings. (**B**–**F**) Statistical data were analyzed using one-way ANOVA and Tukey’s *post-hoc* tests. The statistically significant differences are indicated by a different letter (*p*-value < 0.05) above each column of each histogram.

**Figure 2 plants-10-00130-f002:**
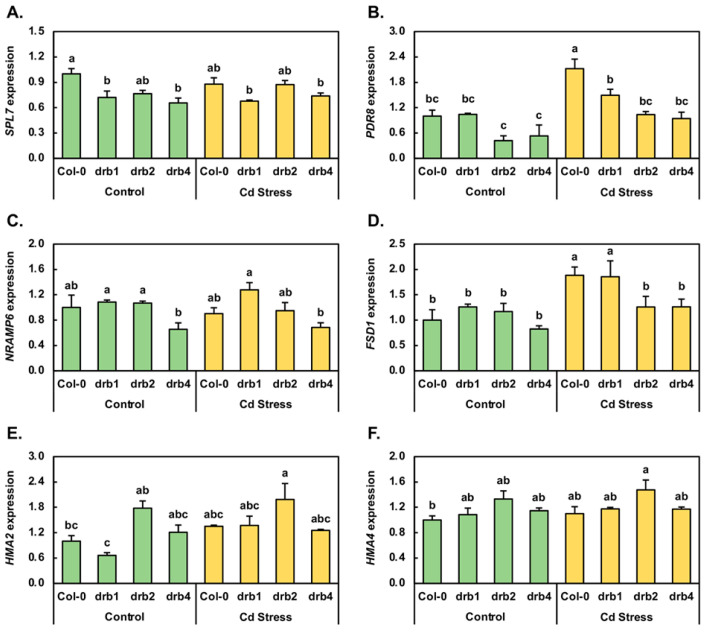
RT-qPCR quantification of the expression of *Arabidopsis* genes known to be responsive to cadmium stress. A standard RT-qPCR approach was used to quantify the transcript abundance of a select group of *Arabidopsis* genes known to be responsive to Cd stress, including (**A**) *Squamosa promoter binding protein-like 7* (*SPL7)*; (**B**) *PDR8*; (**C**) *natural resistance-associated macrophage protein* (*NRAMP6)*; (**D**) *FE superoxide dismutase 1* (*FSD1)*; (**E**) *heavy metal ATPase 2* (*HMA2)* and (**F**) *heavy metal ATPase 4* (*HMA4*). The expression of these six genes was analyzed in 15-day-old Col-0, *drb1*, *drb2* and *drb4* plants following their exposure to a 7-day 50 μM CdCl_2_ stress treatment regime for comparison to the control grown counterpart of each *Arabidopsis* line. (**A**–**F**) Statistical data were analyzed using one-way ANOVA and Tukey’s *post-hoc* tests. The statistically significant differences are indicated by a different letter (*p*-value < 0.05) above each column of each histogram.

**Figure 3 plants-10-00130-f003:**
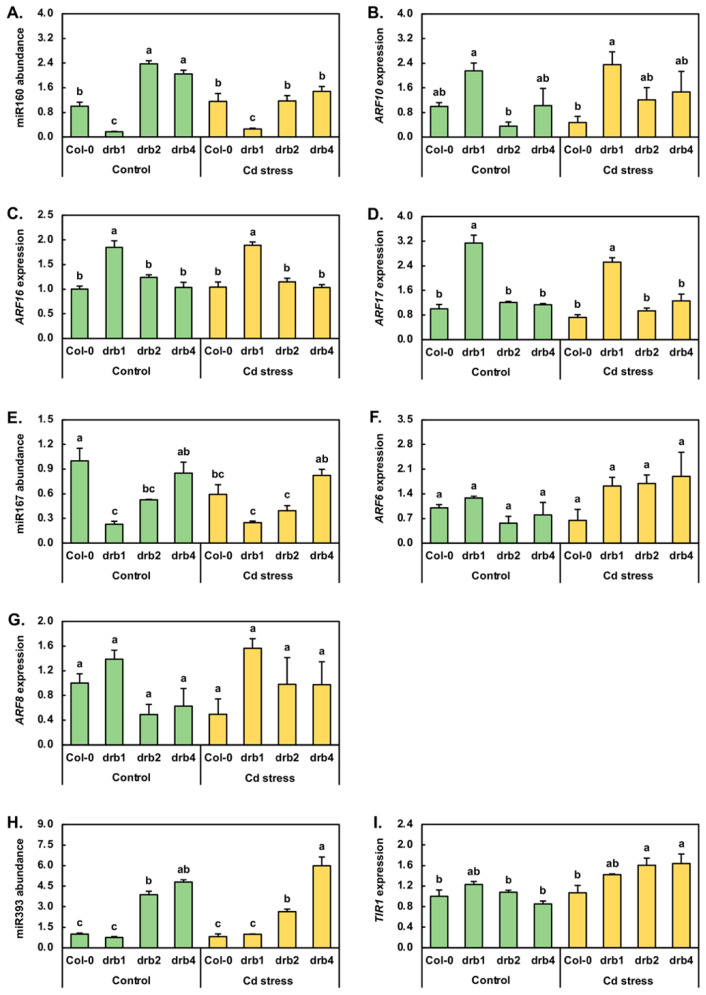
Molecular profiling of the expression modules of three auxin pathway-specific microRNAs following the exposure of Col-0, *drb1*, *drb2* and *drb4* plants to cadmium stress. RT-qPCR quantification of miR160 abundance (**A**) and the expression of its target genes, *ARF10* (**B**), *ARF16* (**C**) and *ARF17* (**D**), in 15-day-old control and Cd-stressed Col-0, *drb1*, *drb2* and *drb4* plants. RT-qPCR quantification of miR167 abundance (**E**) and of the expression of its *ARF* target genes, *ARF6* (**F**) and *ARF8* (**G**), in 15-day-old control and Cd-stressed Col-0, *drb1*, *drb2* and *drb4* plants. Profiling of miR393 abundance (**H**) and of the expression of its primary target gene, *TIR1* (**I**), in control and Cd-stressed Col-0, *drb1*, *drb2* and *drb4* whole seedlings by RT-qPCR. (**A**–**I**) Statistical data were analyzed using one-way ANOVA and Tukey’s *post-hoc* tests. The statistically significant differences are indicated by a different letter (*p*-value < 0.05) above each column of each histogram.

**Figure 4 plants-10-00130-f004:**
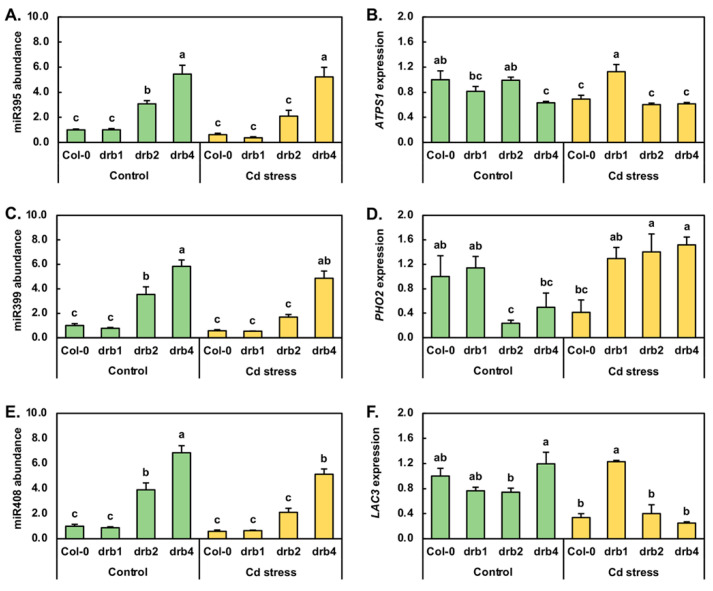
RT-qPCR quantification of the abundance of three *Arabidopsis* abiotic stress responsive microRNAs and the expression of their target genes in cadmium-stressed Col-0, *drb1*, *drb2* and *drb4* seedlings. RT-qPCR assessment of miR395 abundance (**A**) and the expression of its target gene, *ATPS1* (**B**), in control and Cd-stressed Col-0, *drb1*, *drb2* and *drb4* whole seedlings. Quantification of the abundance of miR399 (**C**) and the expression of its target gene *PHO2* (**D**) in Col-0, *drb1*, *drb2* and *drb4* whole seedlings cultivated in either control conditions or a Cd-stress environment. RT-qPCR analysis of miR408 abundance (**E**) and the expression of its target gene, *LAC3* (**F**), in 15-day-old control and Cd-stressed Col-0, *drb1*, *drb2* and *drb4* whole seedlings. (**A**–**F**) Statistical data were analyzed using one-way ANOVA and Tukey’s *post-hoc* tests. The statistically significant differences are indicated by a different letter (*p*-value < 0.05) above each column of each histogram.

**Figure 5 plants-10-00130-f005:**
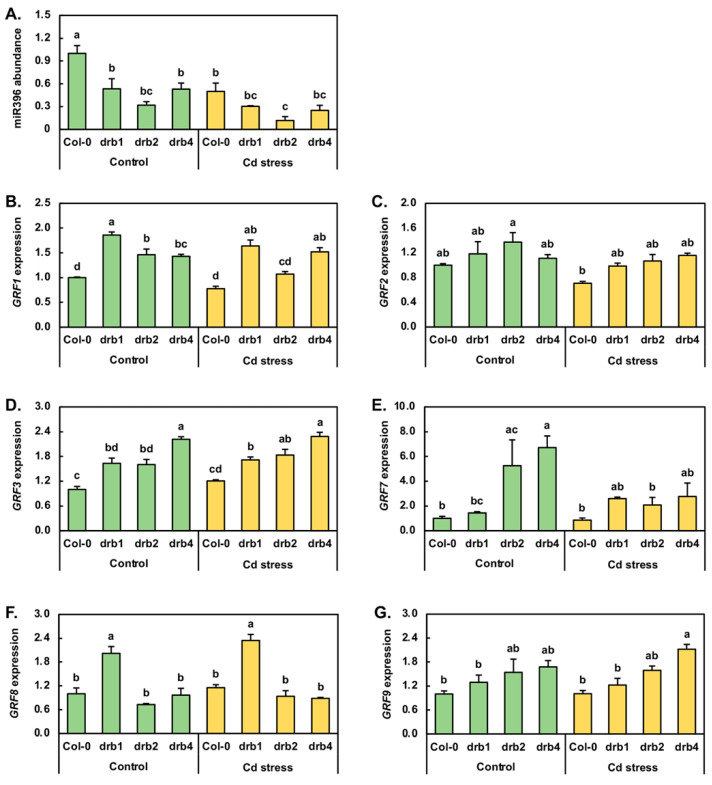
RT-qPCR quantification of the miR396 abundance and *GRF* target gene expression in control and cadmium-stressed Col-0, *drb1*, *drb2* and *drb4* plants. Quantification of miR396 abundance (**A**) and the expression of its *GRF* target genes, *GRF1* (**B**), *GRF2* (**C**), *GRF3* (**D**), *GRF7* (**E**), *GRF8* (**F**) and *GRF9* (**G**), via RT-qPCR assessment of 15-day-old control and Cd-stressed Col-0, *drb1*, *drb2* and *drb4* whole seedlings. (**A**–**G**) Statistical data were analyzed using one-way ANOVA and Tukey’s *post-hoc* tests. The statistically significant differences are indicated by a different letter (*p*-value < 0.05) above each column of each histogram.

## Data Availability

All data reported here is available from the authors upon request.

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
