# Peer review of "MicroRNA-Mediated Responses to Cadmium Stress in *Arabidopsis thaliana"

_plants, 2021, doi:10.3390/plants10010130_

Round 1
Reviewer 1 Report
Authors have attempted in the present study to investigate the complexity of the miRNA directed molecular response to cadmium stress in Arabidopsis. They have assessed the role of miRNA function under cadmium stress treatment by considering drb mutant lines. Although, authors have correlated well the active role of MicroRNA and cadmium stress in Arabidopsis, the following comments still need to be clarified.
- How authors are aware that 50 μM CdCl2 stress is appropriate for this study. Whether there is any chance to get a different result if the concentration of CdCl2 will be changed?
- Similarly, 8+7 day period of seedlings preparation and stress treatment has been selected. How authors are confident that these are the optimum period to induce the response they want to capture and there is no chance of major alteration in results if days change.
Author Response
Dear Reviewer,
The selection of 50 μM CdCl2 stress for a period of 7 days was selected to induce a prolonged, mild-to-moderate phenotypic response in Arabidopsis. The aim of this study was to elucidate the crucial role that miRNAs mediate in underpinning the adaptive response of Arabidopsis to this form of abiotic stress. For this reason, this study was careful to not select a CdCl2 concentration of too high an intensity, basing the selection of 50 μM CdCl2 on previously observed phenotypic responses of Arabidopsis to various doses of CdCl2 (see; https://doi.org/10.1104/pp.014118; https://doi.org/10.1111/jipb.12658).
As with other abiotic stress growth conditions, varied concentrations (intensity) and exposure times to CdCl2 will elicit distinct molecular and phenotypic responses, again see https://doi.org/10.1104/pp.014118 and Figure 4.
- Similarly, 8+7 day period of seedlings preparation and stress treatment has been selected. How authors are confident that these are the optimum period to induce the response they want to capture and there is no chance of major alteration in results if days change.
Dear Reviewer, thank you for this question, and please find below our response;
We selected 8-day-old seedlings to expose to Cd stress as we have previously identified (via non-published preliminary analyses) this seedling age as the appropriate age of Arabidopsis plant development to readily document both the molecular and phenotypic response of this plant species to expose to a range of environmental stresses, including drought, heat, salt and phosphate stress.
In addition, we performed a series of preliminary analysis (not published) to determine what duration of abiotic stress exposure elicited the most readily quantifiable response at both the molecular and phenotypic level to each imposed stress (drought, heat, salt and phosphate stress). Yes – these preliminary analyses did reveal Arabidopsis plant lines of differing age to respond slightly differently to each imposed stress. However, as stated, in our hands, the use of 8-day-old Arabidopsis seedlings combined with a 7-day stress treatment period provided the most readily quantifiable response of the analysed Arabidopsis lines to each assessed stress.
Use of the same age of (1) Arabidopsis seedlings and (2) stress exposure duration further allows for direct comparison between the response of individual Arabidopsis lines to a wide range of abiotic stresses. Please see our previous publications where we have exposed 8-day-old Arabidopsis seedlings to a 7-day drought, heat, salt and phosphate stress exposure treatment period, including (1) Pegler JL et al., (2019a) Plants, Pegler JL et al., (2019b) Plants, Nguyen DQ et al., 2020 Int J Mol Sci, and Pegler JL et al., (2020) Plants.
Reviewer 2 Report
Dear Authors,
I like this interesting paper, the experiment was well designed and conclusion are scientificly justified. Congratulations, can I propose to specify (e.g. in discussion) the standards for cadmium in Australia, Vietnam and the European Union. Did the tested amounts exceed them and if so, how much?
L63 – If you used the pattern English name followed by Latin “…canola (Brassica napus), maize (Zea mays), than better keep it also for others, so change the order e.g. ”…barrelclovers Medicago truncatula and M. sativa(alfalfa)…”
L83 – “RT-qPCR” appears for the first time, please explain! Reverse transcription? Real time? Qualitative PCR
L112-117 – should go to the methodology as description of the experiment
Author Response
Dear Authors,
I like this interesting paper, the experiment was well designed and conclusion are scientificly justified. Congratulations, can I propose to specify (e.g. in discussion) the standards for cadmium in Australia, Vietnam and the European Union. Did the tested amounts exceed them and if so, how much?
Dear Reviewer, the cadmium concentration in the soil of multiple countries (incl. Australia and UK, among many others) is reviewed in https://doi.org/10.1016/j.scitotenv.2017.06.030. The amount of cadmium can vary substantially between geographical locations as contamination stems from an array of factors such as (but not limited to) weathering, industrial practice and vehicular emission. Therefore, the amount/concentration of CdCl2 assessed in this study was not selected on the cadmium standard of a selected location, but rather the ability of the selected concentration of CdCl2 to elicit a prolonged mild-to-moderate phenotypic and molecular response in Arabidopsis to allow for the fundamental study of the miRNA-directed molecular pathways that allow a plant to adapt to, or to tolerate prolonged exposure to this particular form of abiotic stress.
L63 – If you used the pattern English name followed by Latin “…canola (Brassica napus), maize (Zea mays), than better keep it also for others, so change the order e.g. ”…barrelclovers Medicago truncatula and M. sativa(alfalfa)…”
Dear Reviewer: thank you kindly for bringing this oversight to our attention. We have corrected such instances throughout the revised version of our manuscript.
L83 – “RT-qPCR” appears for the first time, please explain! Reverse transcription? Real time? Qualitative PCR
Dear Reviewer: thank you very much for identifying this mistake. We have stated what RT-qPCR stands for (that is; quantitative reverse transcriptase polymerase chain reaction) on its first mention in the text of our revised manuscript for clarity to the reader.
L112-117 – should go to the methodology as description of the experiment.
Dear Reviewer: thank you for this helpful suggestion. We have removed this sentence of text from the Results section of the revised version of our manuscript as we agree with your comment that this sentence belongs in the Materials and Methods section of our manuscript and not in the text of the Results section. Thank you again for your helpful and informative comments / suggestions which have improved the impact of our study.
Reviewer 3 Report
The manuscript presented for review raises the issue of changing the regulation of gene expression towards miRNA caused by a stressful substance.
This work is very interesting and carefully written. It contains a broad discussion study.
The only drawback seems to be the placement of Fig. 1 before the chapter and description of the results presented on it.
I think that this work can be published in its present form after proper arrangement of drawings.
Author Response
Reviewer 3.
Comments and Suggestions for Authors
The manuscript presented for review raises the issue of changing the regulation of gene expression towards miRNA caused by a stressful substance.
This work is very interesting and carefully written. It contains a broad discussion study.
Dear Reviewer: the authors kindly thank you for your positive reviewer of our manuscript.
The only drawback seems to be the placement of Fig. 1 before the chapter and description of the results presented on it.
I think that this work can be published in its present form after proper arrangement of drawings.
Dear Reviewer: we have now changed the placement of the Figures in our revised version of our manuscript in order to place each Figure following the section of text which describes the results reported in each Figure. We thank you for this help suggestion to improve the format, and therefore, the quality of our manuscript in its revised version.